# Inhibition of Plasmid Conjugation in *Escherichia coli* by Targeting *rbsB* Gene Using CRISPRi System

**DOI:** 10.3390/ijms241310585

**Published:** 2023-06-24

**Authors:** Yawen Xiao, Yan Zhang, Fengjun Xie, Rikke Heidemann Olsen, Lei Shi, Lili Li

**Affiliations:** 1Institute of Food Safety and Nutrition, Jinan University, Guangzhou 510632, China; yawen929@stu2018.jnu.edu.cn (Y.X.); fgwantse@163.com (F.X.); leishi@jnu.edu.cn (L.S.); 2Department of Veterinary and Animal Sciences, Faculty of Health and Medical Sciences, University of Copenhagen, 1870 Frederiksberg, Denmark; cava@sund.ku.dk

**Keywords:** CRISPRi, *rbsB*, conjugation, biofilm, quorum sensing

## Abstract

Bacterial conjugation constitutes a major horizontal gene transfer mechanism for the dissemination of antibiotic-resistant genes (ARGs) among human pathogens. The spread of ARGs can be halted or diminished by interfering with the conjugation process. In this study, we explored the possibility of using an *rbsB* gene as a single target to inhibit plasmid-mediated horizontal gene transfer in *Escherichia coli* by CRISPR interference (CRISPRi) system. Three single-guide RNAs (sgRNAs) were designed to target the *rbsB* gene. The transcriptional levels of the *rbsB* gene, the conjugation-related genes, and the conjugation efficiency in the CRISPRi strain were tested. We further explored the effect of the repressed expression of the *rbsB* gene on the quorum sensing (QS) system and biofilm formation. The results showed that the constructed CRISPRi system was effective in repressing the transcriptional level of the *rbsB* gene at a rate of 66.4%. The repressed expression of the *rbsB* gene resulted in the reduced conjugation rate of RP4 plasmid by 88.7%, which significantly inhibited the expression of the conjugation-related genes (*trbBp*, *trfAp*, *traF* and *traJ*) and increased the global regulator genes (*korA*, *korB* and *trbA*). The repressed *rbsB* gene expression reduced the depletion of autoinducer 2 signals (AI-2) by 12.8% and biofilm formation by a rate of 68.2%. The results of this study indicated the *rbsB* gene could be used as a universal target for the inhibition of conjugation. The constructed conjugative CRISPRi system has the potential to be used in ARG high-risk areas.

## 1. Introduction

The antibiotic resistance of bacteria (ARB) has been recognized as a serious threat to public health by the World Health Organization (WHO) (https://www.who.int/news-room/fact-sheets/detail/antibiotic-resistance, accessed on 1 January 2019.). There is an urgent need to develop effective new strategies to combat ARB [1,2]. Antibiotic resistance genes (ARGs) can easily spread to bacterial pathogens by horizontal gene transfer (HGT) [3,4]. Plasmid conjugation is one of the main sources of HGT, and the emergence of multi-drug resistant (MDR) pathogens is frequently linked to the spread of conjugative plasmids [5,6]. Preventing plasmid conjugation is therefore a key step in curbing the propagation of ARGs.

Traditional strategies for curing plasmids involve treating their growth at elevated temperatures, utilizing UV light, or adding chemical agents (e.g., SDS, ethidium bromide or acridine orange, unsaturated fatty acids, 2-hexadecynoic acid, and synthetic 2-alkynoic fatty acids) [7,8,9] to interfere with plasmid replication by integrating modified bases into the DNA, causing breaks in the DNA, or eliciting other mechanisms [10]. However, these compounds incur the risk of generating adverse and unwanted mutations in the bacterial chromosome. Moreover, these methods lack specificity to target a particular type of plasmid for bacteria containing different types of plasmids [10]. Furthermore, some of these compounds have stability, toxicity or scarcity problems that need to be addressed [10]. Other ways for plasmid-curing include a molecular biology tool based on replicon incompatibility between identical replicons [11] and utilizing pCURE plasmid displacement, which involves combining key regions of replicons and the post-segregational killing systems for IncP-1 and IncF plasmid curing [12]. The advantages of these methods include the low risk of chromosomal mutations and the high specificity to the target plasmid. However, they require precise knowledge of the replication machinery of the target plasmid. Therefore, alternative effective and simple methods for the inhibition of plasmid conjugation are needed to combat bacterial MDR development.

Recently, the clustered regularly interspaced short palindromic repeats (CRISPR) RNA-guided Cas9 (CRISPR-Cas9) system has been applied as a method for plasmid curing [10,13,14]. The CRISPR/Cas9-based plasmid-curing system requires only the coexpression of a Cas protein and a customizable single-guide RNA (sgRNA). The CRISPR/Cas9 system has been harnessed to cure plasmid by targeting specific ARGs or replication-related genes, transfer-related genes, toxin–antitoxin system genes on plasmid [13,15,16]. Derived from CRISPR-Cas9, the CRISPR interference (CRISPRi) system enables the rapid and efficient silencing of genes without altering the target DNA sequence. This system has been widely used to identify functional genes, suppress antibiotic resistance, and regulate microbial metabolism [17,18,19,20,21]. The CRISPRi system is inducible and fully reversible, effectively targeting specific strains without disrupting the entire microbiome and local environment. 

The ribose-binding protein RbsB in *Escherichia coli* is a periplasmic binding protein that participates in the high-affinity membrane transport process, and a subset also serves as the primary chemoreceptor for chemotaxis [22,23]. Apart from this known function, some studies have indicated that RbsB might play an important role in the conjugation and quorum-sensing (QS) system [24,25,26]. As the *rbsB* gene is ubiquitous in bacteria and has a potential role in mediating conjugation, it might be used as a universal target to inhibit plasmid conjugation. Thus, in this study, we explored the possibility of targeting the *rbsB* gene by engineering a CRISPRi system to inhibit plasmid conjugation and further analyzed its effects on the QS system and biofilm formation to provide insights for developing methods for conjugation inhibition.

## 2. Results

### 2.1. Titration of CRISPRi System

To determine the appropriate concentration of anhydrotetracycline (aTc) that sufficiently triggered the CRISPRi system without affecting its bacterial growth, we evaluated the growth and dCas9 transcriptional level of the control strain *E. coli* HB101 (RP4+plv-dCas9-B0) in serial dilutions of aTc. The plasmid plv-dCas9-B0 contained a sgRNA (B0) targeting sequence that is not homologous to any *E. coli* HB101 sequence (Table 1). The results revealed no difference in the growth rates between the CRISPRi strains and the control strain without aTc exposure (Figure 1A), suggesting that the CRISPRi system has no significant effect on *E. coli* HB101 (RP4) growth and that the leakage of gene expression could be avoided by strictly controlling the P_tetO_ promoter. The growth of *E. coli* HB101 (RP4+plv-dCas9-B0) was slightly compromised in aTc concentrations up to 2 μM and 4 μM. However, the addition of 0.25 μM, 0.5 μM, and 1 μM of aTc demonstrated no significant effect on the growth of the strain (Figure 1B). RT-qPCR showed that dCas9 transcription relied on aTc induction, and its mRNA level peaked when the aTc concentration reached 1 µM (Figure 1C). Thus, an appropriate aTc concentration of 1 μM was selected for triggering the CRISPRi system in the following study.

Upon aTc induction, the CRISPRi system with sgRNAs (B1) showed a significant repression effect (*p* < 0.0001) on the *rbsB* gene, at a rate of 66.4% (Figure 1D). The *rbsC* and *rbsK* genes within the *rbsDACBK* operon were also significantly repressed compared with the control (Appendix A). No significant inhibitory effect was observed on sgRNAs (B2) and sgRNAs (B3). The expression level of the *rbsB* gene in a wild type (WT) strain showed no significant difference from the control strain, indicating that the repression of the *rbsB* gene was not affected by the transformation procedure and plasmids (Figure 1D). Thus, the CRISPRi system with sgRNAs (B1) was selected in the following studies.

### 2.2. Effect of CRISPRi System on Conjugation Transfer

Upon aTc induction, only the CRISPRi strain with sgRNAs (B1) significantly reduced the conjugation rate of plasmid RP4 by 88.7% (Figure 2A). No significant inhibitory effect was observed on sgRNAs (B2) and sgRNAs (B3). Accordingly, the mRNA expression levels of conjugation-associated genes (*trbBp*, *trfAp*, *traF* and *traJ*) were decreased significantly by 35.5%, 31.1%, 19.1% and 32.9%, respectively. The global regulator genes (*korA*, *korB* and *trbA*) were increased significantly by 26.3%, 21.7% and 22.0%, respectively (Figure 2B).

### 2.3. Effect of CRISPRi System on QS

Compared with the control strain, AI-2 production was higher by 12.8% in the CRISPRi strain with sgRNAs (B1) during the late exponential and early stationary phases (Figure 3A). At the stationary phase, the level of AI-2 in a supernatant harvested from cultures of both the parental strain and the CRISPRi strain with sgRNAs (B1) decreased. Inhibition of the expression level of the *rbsB* gene influenced the rate of depletion of AI-2 but did not completely inhibit the ability of the CRISPRi strain with sgRNAs (B1) to deplete AI-2 from the solution. Notably, the mRNA expression level of the QS-related gene (*luxS*) showed no significant difference between the CRISPRi strain and the control (Figure 3B).

### 2.4. Effect of CRISPRi on Biofilm Formation

Compared with control, the aTc-induced CRISPRi strain with sgRNAs (B1) significantly reduced the biofilm formation, at a rate of 68.2% (Figure 4).

## 3. Discussion

Conjugation is one of the most important means in the dissemination of antibiotic resistance and virulence factors among pathogenic bacteria [29,30]. By repression of the expression of genes related to conjugation, the transmission of plasmids can be inhibited [29,31]. A previous study has reported that RbsB might play key roles in the transmission of conjugation plasmids [24]. Also observed in our previous study, the reduced conjugation rate was associated with a down-regulated expression of the *rbsB* gene [32]. However, the modulation mechanism of RbsB on conjugation has not been fully elucidated. Hence, we planned to investigate whether plasmid transfer in *E. coli* can be inhibited by targeting the *rbsB* gene, thus further elucidating the modulation mechanism.

As CRISPRi systems have been applied to repress t gene expression with a single plasmid [17,19,20,28,33], in this study, we developed a CRISPRi system targeting a single gene, *rbsB*, to inhibit the conjugation in *E. coli*. The screened CRISPRi strain with sgRNAs (B1) we designed was found to be able to repress the expression of the target gene *rbsB* without affecting the growth of the strain and was capable of decreasing the conjugation rate of plasmid RP4 between *E. coli* strains. As expected, *the rbsC* and *rbsK* genes upstream and downstream of the *rbsB* gene were also repressed, as CRISPRi has been shown to impose a polar effect on upstream and downstream genes of the target gene in operon [34]. The results of this study suggest the *rbsDACBK* operon was associated with plasmid conjugation, and RbsB can be used as a potential target to inhibit plasmid transfer.

The process of conjugation can be affected and regulated by many factors, such as the Mating pair formation (Mpf) system encoding a type IV secretion system (T4SS), which form the conjugative pore; DNA transfer and replication (Dtr) genes-encoded relaxosome composed of the relaxase, which nicks at the origin of transfer (oriT) and other auxiliary proteins; and the global regulator genes, which contribute significantly to activating the Mpf system [35,36,37,38]. For plasmid RP4, the expression of Mpf genes (*trbBp* and *traF*) and Dtr system genes (*trfAp* and *traJ*) was positively related to the formation of conjugants, and the global regulator genes (*korA*, *korB* and *trbA*) were negatively associated with conjugation-associated genes (*trbBp*, *trfAp*, *traF* and *traJ*) [31,35,39,40,41]. The mRNA expression levels of the global regulator genes (*korA*, *korB* and *trbA* genes) were found to be significantly increased when the expression of the *rbsB* gene was repressed, which in turn significantly reduced the expression of conjugation-associated genes (*trbBp*, *trfAp*, *traF* and *traJ*), thereby decreasing the conjugation frequency. These findings suggest RbsB can modulate Mpf and Dtr systems as well as the expression of the conjugation of global regulator genes.

RbsB has been confirmed to mediate quorum signal (AI-2) uptake in nontypeable *Haemophilus influenzae* and *Actinobacillus actinomycetemcomitans* strains [25,26]. In Armbruster’s study, RbsB was identified as a LuxS/AI-2-regulated protein that was required for the uptake of and response to AI-2 in the *H. influenzae* strain. AI-2 was significantly accumulated in supernatant samples in the *rbsB* mutant *H. influenzae* strain during the late-exponential and early-stationary phases [25]. Similarly, AI-2 uptake was significantly less in the CRISPRi strain with sgRNA (B1) than in the control strain at the same time in this study, indicating that the repression of the *rbsB* gene expression also affected the QS between *E. coli* strains. Thus, the decreased conjugation rate by inhibiting the expression of the *rbsB* gene may partly result as the deceased QS between *E. coli* strains. Novel mechanisms for the detection of and response to AI-2 have been identified as widely existing among prokaryotic species [42]. Therefore, it is necessary to further investigate the relationship of RbsB, QS, and plasmid conjugation to find effective strategy to combat the development of antibiotic resistance.

Biofilm formation plays a substantial role in the transfer and dissemination of conjugative plasmids [43,44]. Conjugative transfer was shown to be considerably higher in biofilms [45,46]. RbsB has been reported to be linked to biofilm formation in *E. coli* [47]. In this study, the repression of the expression of *rbsB* gene was found to lead to significantly reduced biofilm formation in the CRISPRi strain with sgRNA (B1). Thus, the repression of the expression of the *rbsB* gene caused attenuated biofilm formation, which might also be a contributor to the reduced conjugation rate.

Furthermore, the results of this study indicated that the constructed CRISPRi system targeting the *rbsB* gene is a simple and efficient method to inhibit plasmid conjugation. The constructed plasmid vector was easily provided with different sgRNAs to target different genes. Moreover, the vector was readily transferred into *E. coli* strains by natural conjugation, indicating it has a great potential to be applied in ARG high-risk areas, such as farms where livestock are often exposed to antibiotics, manure, wastewater treatment, downstream of pharmaceutical (antibiotic) factories, and aquaculture to reduce drug resistance.

## 4. Materials and Methods

### 4.1. Bacterial Strains, Plasmids and Culture Conditions

The bacterial strains and plasmids utilized in this study are listed in Table 1. All *E. coli* strains were cultured in LB broth or agar plates (Guangdong Huankai Microbial Sci &Tech, Guangzhou, China), supplemented with antibiotics when appropriate. For induction of single-guide RNA (sgRNA) and dCas9 expression, aTc was used.

### 4.2. Construction of CRISPRi Vectors

Vector plv-dCas9-sgRNA was utilized as the backbone of the CRISPRi system [28]. This backbone contains an inactive *dCas9* gene from *Streptococcus pyogenes* and a sgRNA chimera, both of which can be expressed under the TetR-inducible P_tetO_ promoter. The sgRNA chimera contains three parts: the base-pairing region (BPR) containing 20 bp of DNA complementary to the target sequence; the dCas9 handle (DH), which is a 42 bp hairpin region for dCas9 binding; and *rrnB* (Ter), which is a 40 bp terminator (Figure 5A) [20]. To construct new CRISPRi recombinant plasmids, only the sgRNA sequence in vector plv-dCas9-sgRNA needed to be replaced (Figure 5C).

Because a non-template strand has been confirmed to be a more effective target for sgRNA than the template strand [20,48], we therefore designed three candidate sgRNAs (B1-B3) directly targeting the non-template strand of the *rbsB* gene (Figure 5B). The target locus was near the start codons and downstream of CCN (N represents A, T, G or C) in coding sequences (CDs) of *rbsB* gene (Figure 5B). The specificity of the sgRNAs was examined by a BLAST search. A sgRNA targeting no sequences in any of the experimental strains was prepared as the control. All oligonucleotides used to construct recombinant plasmids are listed in Table 1 and Appendix A.

Briefly, two complementary oligonucleotides containing about 20 bases homologous to the target sequence plus 3 bases at the 5′ end of each oligonucleotide matching the BspQI-digested vector were synthesized, annealed, phosphorylated, and ligated into plv-dCas9-sgRNA to form the desired CRISPRi recombinant plasmids. All recombinant plasmids were individually transformed into competent *E. coli* HB101 (RP4) cells and confirmed by colony PCR and sequencing. The recombinant strains obtained are shown in Table 1.

### 4.3. Determination of Bacterial Growth Curves

Overnight cultures of the recombinant strains and the control strain were diluted to OD_600_ of 0.01. Cultures were further incubated in a 100-well plate containing 25 mg/L chloramphenicol (Chl), 40 mg/L kanamycin (Km) and 2-fold serial dilutions of aTc from 0.125 to 4 μM to induce the expression of dCas9 and sgRNA. The leakage of gene expression in the CRISPRi strains was tested by incubating CRISPRi strains in an LB broth without adding aTc. The strains were cultivated at 37 °C for 48 h and the cell concentrations were measured every 30 min utilizing a Bioscreen instrument (Lab Systems Helsinki, Finland).

### 4.4. Conjugation Transfer Experiments

To construct the donor strains, the recombinant plasmids plv-dCas9-sgRNA (B0-B3) were transferred to the host strain *E. coli* TOP10 individually, and then transferred to *E. coli* HB101 containing plasmid RP4 by natural conjugation, respectively. The donor *E. coli* HB101 strains (given by Prof. Junwen Li, Institute of Health and Environmental Medicine, Tianjin, China), carried a broad-host-range plasmid RP4 with Km, ampicillin (Amp), and tetracycline (Tc) resistant genes and recombinant plasmids plv-dCas9-sgRNA (B0-B3). *E. coli* J53 with sodium azide (Na_3_N) resistance was the recipient.

The conjugation experiments were performed as described previously [49], with minor modifications. Briefly, the strains were cultured overnight in an LB broth with corresponding antibiotics added (Km: 40 mg/L, Amp: 50 mg/L, Tc: 60 mg/L; Na_3_N: 150 mg/L). The cells were washed three times with PBS buffer solution and the concentrations of donor and recipient strains were adjusted to 10^7^ CFU/mL and mixed in a 1:1 ratio, and aTc was added to induce the CRISPRi system. After incubating at 37 °C for 10 h, diluting the bacterial solution and using the plate count method to count the number of recipients (resistant to Na_3_N) and transconjugants (resistant to Km, Amp, Tc and Na_3_N), the conjugative transfer rate was calculated according to the following formula: Conjugative transfer rate = number of transconjugants (CFU/mL)/number of recipients (CFU/mL).

### 4.5. Reverse Transcription Qualitative PCR (RT-qPCR)

RT–qPCR was performed to quantify the genes expression, including *rbsB* gene and other genes in *rbsDACBK* operon, QS related gene (*luxS*), conjugation-associated genes (*trbBp*, *trfAp*, *traF* and *traJ*) and horizontal transfer global regulator genes (*korA*, *korB* and *trbA*). The16S rRNA gene was used as an internal control. Primers were designed by Primer Premier version 6.0 software or as previously described [50]. Primers of the target genes are listed in Appendix A.

To quantify the *rbsB* gene, the recombinant *E. coli* HB101 strains added with or without aTc were grown in an LB broth at 37 °C to the logarithmic phase. The total RNA was extracted, using the RNA extraction kit (Magen, Guangzhou, China), and reverse transcription was conducted immediately, using the RT ProMix kit (CISTRO, Guangzhou, China) following the manufacturer’s instructions.

To quantify the QS-related gene (*luxS*), conjugation-associated genes (*trbBp*, *trfAp*, *traF* and *traJ*) and horizontal transfer global regulator genes (*korA*, *korB* and *trbA*), conjugative transfer was performed for 10 h at 37 °C upon aTc induction. Then, the bacterial cell pellets in the conjugation system were collected by centrifugation (10,000 rpm for 5 min). RNA extraction and reverse transcription were carried out as described above.

RT-PCR was performed with SYBR Green Pro Taq HS kit (AG, Guangzhou, China) by using a QuantStudio^TM^ 6 Flex System (Thermo Fisher Scientific, Waltham, MA, USA). The reverse transcription reaction was performed with 100 ng of RNA in a 10 μL final reaction volume. All the RT–qPCR assays were repeated three times, and each experiment was performed in triplicate. The qPCR data were analyzed using the 2^−ΔΔCT^ method (where CT is the threshold cycle) with *E. coli* 16S rRNA as an internal control for normalization.

### 4.6. Detection of AI-2 Signaling Molecule

To investigate the effect of repression of the *rbsB* gene expression on the QS system in *E. coli*, an autoinducer 2-signals (AI-2) bioluminescence assay was performed as described previously [51], with minor modifications. Briefly, bacterial cultures added with or without 1 μM aTc were grown to the stationary phase. Samples were taken at 2 h intervals and centrifuged at 12,000g for 10 min, and the cell-free culture fluid was collected by filtering the supernatant with a 0.22 μm filter (Millipore, Bedford, MA, USA). The AI-2 reporter strain *Vibrio harveyi* BB170 was incubated in an AB medium overnight at 30 °C, and then the culture was diluted 1:5000 in a fresh AB medium. Then the cell-free culture fluid and the diluted *V. harveyi* BB170 culture were mixed at the ratio of 1:9 and incubated at 30 °C for 5 h. During incubation, 200 μL aliquots were transferred to the 96-well plate and the bioluminescence was measured using a Multimode Plate Reader (Tecan, Infinite M200, Mannedorf, Switzerland) at a 1 h interval. The cell-free culture fluid from *V. harveyi* BB170 was used as the positive control; the cell-free culture fluid from *E. coli* DH5α was used as the negative control, and AB medium was used as the blank control. The bioluminescence of all samples was recorded immediately when the blank control reached the lowest value. The relative activity of AI-2 in the cell-free culture fluid was expressed as a percentage of the positive control. All the assays were repeated three times, and each experiment was performed in triplicate.

### 4.7. Biofilm Formation Assay

To determine the effect of repression of the *rbsB* gene expression on biofilm formation, the latter was assessed using the crystal violet staining method described previously [52], with minor adjustments. Briefly, overnight cultures of the recombinant *E. coli* HB101 strains were diluted to a ratio of 1:250 in fresh TSB medium (with 0.2% glucose) supplemented with 25 mg/L Chl, 40 mg/L km with or without 1 μM aTc. The cultures were cultured in a 96-well plate (200 μL in each well) at 37 °C for 24 h without shaking. After incubation, the medium was carefully removed, and the plate was gently rinsed with PBS (1×) solution to remove planktonic cells. Then, the biofilms in the wells were fixed at 42 °C for 30 min. The wells were stained with 200 μL of 0.1% crystal violet solution at room temperature for 15 min and washed with PBS (1×) solution. Then, 100 μL of 95% absolute alcohol were added to each well and the plates were kept on a shaker for 10 min to release the dye properly. The biofilm formation was quantified by measuring the OD590 of the suspension using a microplate reader (Tecan, Infinite F50 microplate reader, Mannedorf, Switzerland).

### 4.8. Statistical Analysis

All experiments were conducted in triplicate. Statistical analyses were performed in GraphPad Prism version 8.0.1 (San Diego, CA, USA). Data were presented as mean ± standard error of mean (SEM) and differences between mean values were tested via *t*-test and one-way analysis of variance (ANOVA). Differences were compared at confidence levels of *p* < 0.05, *p* < 0.01, *p* < 0.001 and *p* < 0.0001.

## 5. Conclusions

To summarize, this study confirmed that the *rbsB* gene can be used as a universal target to inhibit plasmid conjugation in *E. coli*. The repressed expression of the *rbsB* gene reduced the plasmid conjugation rate between *E. coli* strains and inhibited the expression of conjugation-related genes, QS and biofilm formation. Moreover, the results of this study indicated the conjugative CRISPRi system has potential to be used in ARG high-risk areas to prevent the rise and persistence of antibiotic-resistant bacteria.

## Figures and Tables

**Figure 1 ijms-24-10585-f001:**
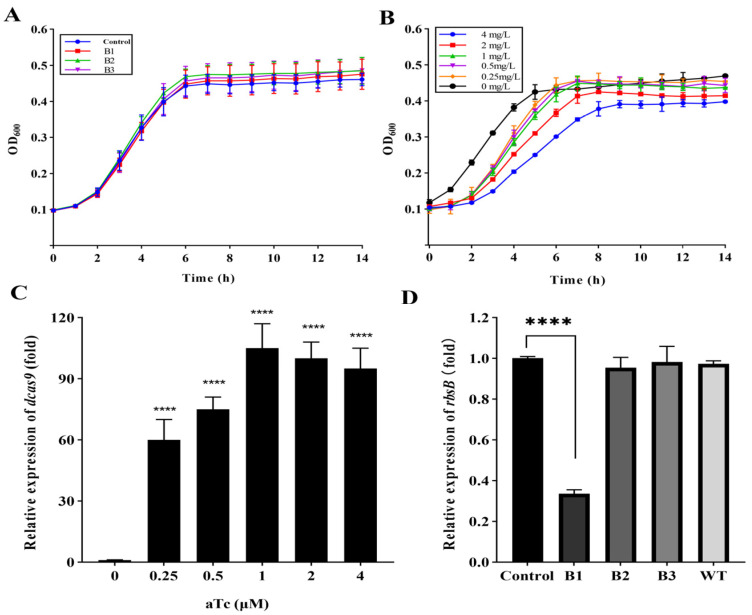
Titration of the CRISPRi system in *E. coli*. (**A**) Growth curves of the CRISPRi strains without induction of the CRISPRi system. (**B**) Growth curve of the control strain upon induction with 2-fold serial concentrations of aTc ranging from 0 to 4 μM. (**C**) Transcription of the *dcas9* gene in the control strain *E. coli* HB101 (RP4+plv-dCas9-B0) upon induction by aTc ranging from 0 to 4 μM. (**D**) Transcription of the *rbsB* gene in recombinant *E. coli* strains upon induction by aTc. B0, B1, B2 and B3 refer to the control strain, strains with plv-dCas9-B1, plv-dCas9-B2 and plv-dCas9-B3, respectively. WT refers to the wild type *E. coli* HB101 strain. Asterisks indicate significant differences between the results for control and B1 strain (****, *p* < 0.0001). All data represent the means ± standard error of mean of biological triplicates.

**Figure 2 ijms-24-10585-f002:**
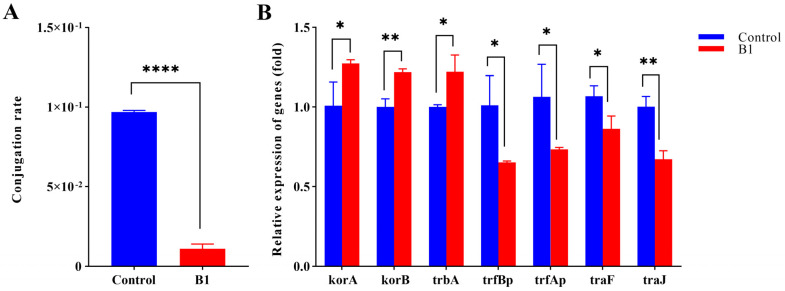
Effect of the CRISPRi system on the plasmid RP4 conjugation rate (**A**) and the mRNA expression level of global regulator genes (*korA*, *korB* and *trbA*) and conjugation-relevant genes (*trbBp*, *traF*, *trfAp* and *traJ*) (**B**). Control refers to *E. coli* HB101 (RP4+plv−dCas9−B0), and B1 refers to the strain with plv−dCas9-B1. All data represent the means ± standard error of mean of biological triplicates. Asterisks indicate significant differences between results for the control and B1 strain (*, *p* < 0.05; **, *p* < 0.01; ****, *p* < 0.0001).

**Figure 3 ijms-24-10585-f003:**
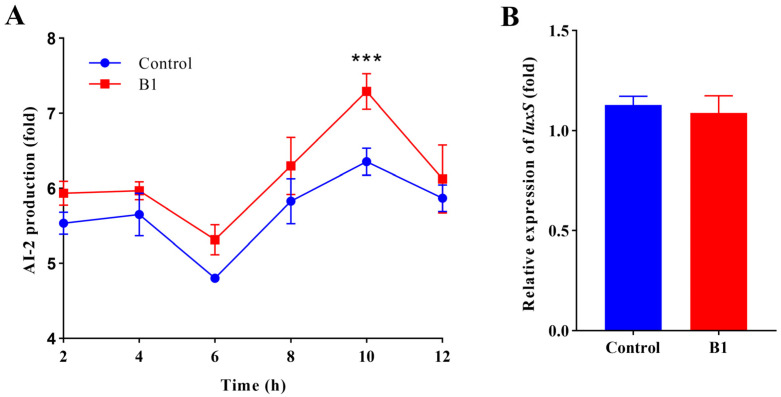
Effect of the CRISPRi system on the quorum sensing (QS) system. (**A**) Measurement of AI-2 activity using bioluminescence assay. The AI-2 activity was expressed as the ratio of bioluminescence of the test isolate to that of *V. harveyi* BB170 (positive control). The values were the means of three independent experiments. (**B**) The mRNA expression level of the QS-related gene (*luxS*). Control refers to *E. coli* HB101 (RP4+plv-dCas9-B0), and B1 refers to the strain with plv-dCas9-B1. Error bars indicate standard deviations. (***, *p* < 0.001).

**Figure 4 ijms-24-10585-f004:**
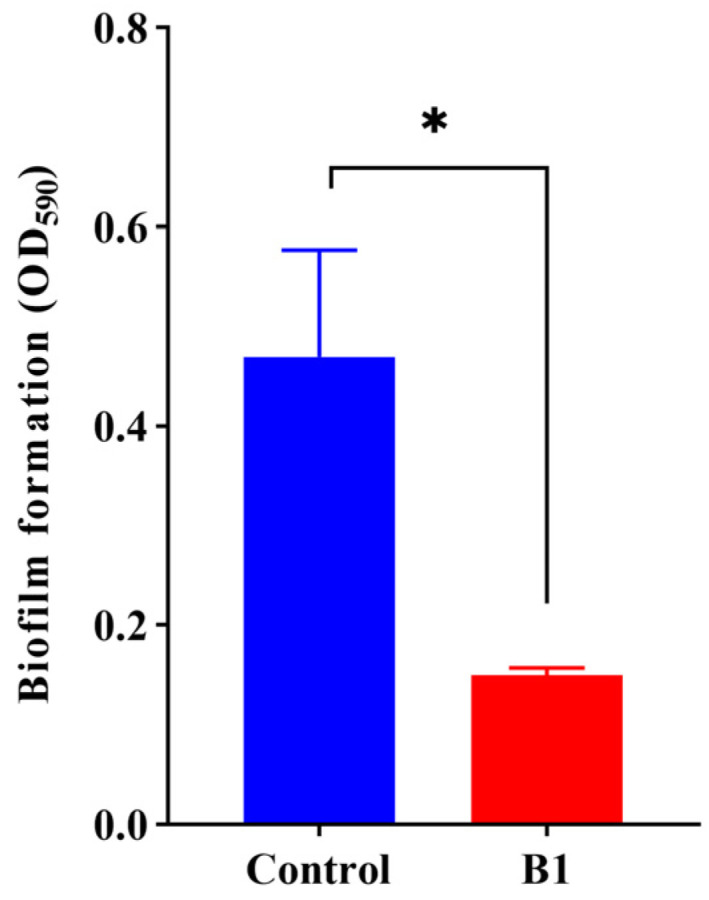
Effect of the CRISPRi system on biofilm formation. Control refers to *E. coli* HB101 (RP4+plv-dCas9-B0), and B1 refers to the strain with plv-dCas9-B1. All data represent the means ± standard error of mean of biological triplicates. Asterisks indicate significant differences between results for control and B1 strain (*, *p* < 0.05).

**Figure 5 ijms-24-10585-f005:**
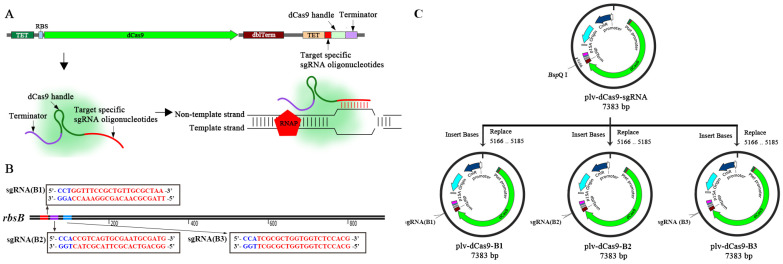
Schematic diagram of engineering a CRISPRi system to repress *rbsB* gene expression in *E. coli*. (**A**) Harnessing CRISPRi to block transcription. RNAP, RNA polymerase. (**B**) Selection of target gene sequence (red color) based on protospacer adjacent motif (PAM) location (blue color), GC content, RNA secondary structure. (**C**) Protocol for construction of CRISPRi recombinant plasmids.

**Table 1 ijms-24-10585-t001:** Strains and plasmids utilized in this study.

Plasmids/Strains	Description ^a^	Reference/Source
Plasmids		
RP4	Km^R^, Amp^R^, Tc^R^	[27]
plv-dCas9-sgRNA	Backbone	[28]
plv-dCas9-B0	Containing CRISPRi system with sgRNA(B0) targeting sequence that is not homologous to any *E. coli* HB101 sequence, Cm^R^	This study
plv-dCas9-B1	plv-dCas9-sgRNA plasmid containing sgRNA(B1), Cm^R^	This study
plv-dCas9-B2	plv-dCas9-sgRNA plasmid containing sgRNA(B2), Cm^R^	This study
plv-dCas9-B3	plv-dCas9-sgRNA plasmid containing sgRNA(B3), Cm^R^	This study
Strains		
*E. coli* TOP10	Host stain of vector plv-dCas9-sgRNA	This study
*E. coli* J53	Na_3_N^R^, used as recipient	This study
*E. coli* HB101 (RP4)	*E. coli* HB101 carrying plasmid RP4	[27]
*E. coli* HB101 (RP4+plv-dCas9-B0)	*E. coli* HB101 (RP4) carrying plasmid plv-dCas9-sgRNA (B0), used as control	This study
*E. coli* HB101 (RP4+plv-dCas9-B1)	*E. coli* HB101 (RP4) carrying plasmid plv-dCas9-sgRNA (B1), used as donor	This study
*E. coli* HB101 (RP4+plv-dCas9-B2)	*E. coli* HB101 (RP4) carrying plasmid plv-dCas9-sgRNA (B2), used as donor	This study
*E. coli* HB101 (RP4+plv-dCas9-B3)	*E. coli* HB101 (RP4) carrying plasmid plv-dCas9-sgRNA (B3), used as donor	This study

^a^ Km^R^, kanamycin resistant; Amp^R^, ampicillin resistant; Tc^R^, tetracycline resistant; Cm^R^, chloramphenicol resistant.

## Data Availability

Data are contained within the article or Appendix A.

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
