# Peer review of "Inhibition of Plasmid Conjugation in Escherichia coli by Targeting rbsB Gene Using CRISPRi System"

_ijms, 2023, doi:10.3390/ijms241310585_

Round 1

Reviewer 1 Report

In this manuscript, the authors investigated the influence of rpsB repression to plasmid conjugation efficiency in E. coli. CRISPRi system was used to reduce expression level of rpsB. The authors showed that reduced mRNA level of rpsB leads to decreasing the conjugation rate of plasmid RP4 between E. coli strains as well as changing expression level of conjugation related genes (trbBp, trfAp, traF, and traJ) and the global regulator genes (korA, korB and trbA).

Main comments:

-        I would like to know how authors can explain such small fold-repression rpsB through CRISPRi. This gene is non-essential in E. coli [1] and CRISPRi system can effectively silence target genes hundreds-fold [2].

-        It has been shown that CRISPRi imposes a polar effect on upstream and downstream genes of target gene in operon [3]. According to previous data [4], [5] rpsB is а part of rbsDACBK operon. Without checking the level of expression of rbsС and rbsK, it can’t be declared that the effects observed in this research are associated with the function of rpsB.

-        The authors rightly noted that process of conjugation can be affected by many factors. Moreover, there is a scanty knowledge about role of RbsB on plasmid conjugation. High throughput analysis (like RNA-seq or proteomic analysis) could help to compose molecular profile of control and rpsB-downregulated strains. In this research only 8 genes were tested. The results of this study are very limited. 

[1] Goodall ECA, et al. The Essential Genome of Escherichia coli K-12. mBio. 2018

[2] Qi L.S., et al, Repurposing CRISPR as an RNA-guided platform for sequence-specific control of gene expression. Cell. 2013

[3] Peters, J.M. et al. A comprehensive, CRISPR-based functional analysis of essential genes in bacteria. Cell 2016

[4] Shimada T. et al, Involvement of the ribose operon repressor RbsR in regulation of purine nucleotide synthesis in Escherichia coli, FEMS Microbiology Letters, 2013

[5] Laikova O.N. et al, Computational analysis of the transcriptional regulation of pentose utilization systems in the gamma subdivision of Proteobacteria, FEMS Microbiology Letters, 2001

Minor comments:

-        Line 81-82,

The start of Results section confuses the reader. Please add some explanations what is plv-dCas9-101-B0, B1 and etc

-        Figure 1,

Please unify the name of the control sample (B0 or control) in the Figure 1.

-        Line 87-92, Figure 1 B,

“The growth of E. coli 87 HB101 (RP4+plv-dCas9-B0) were slightly compromised in aTc concentrations up to 2 μM and 4 μM. While, the addition of 0.25 μM, 0.5 μM and 1 μM of aTc have no significant effect on the growth of the strain (Figure 1B). The dCas9 gene transcription relied on aTc induction. Thus, an appropriate aTc concentration of 1 μM was selected for triggering the CRISPRi system in the following study.”

I wonder why the authors didn’t measure the level of dcas9. If you titrate concentration of inductor, you also should measure the level of induction of target gene

-        Figure 1C,

I recommend to add the expression level of rpsB in WT in order to exclude possible influence of transformation procedure and plasmid on rpsB expression.  

-        Figure 2,

Please clarify what is Control sample in this experiment

Author Response

Dear reviewer,

 We wish to thank you for your effort in reviewing our manuscript and giving us an opportunity to revise our manuscript. We are also very grateful to your valuable comments in order to improve the quality of our manuscript. We have carefully reviewed and revised our manuscript according to your opinions and comments. All the revised parts are marked in red in the revised manuscript. The responses to the comments are as follows.

Main comments:

-        I would like to know how authors can explain such small fold-repression rpsB through CRISPRi. This gene is non-essential in E. coli [1] and CRISPRi system can effectively silence target genes hundreds-fold [2].

Response: Thank you very much for your comments. In this study, we focused on exploring the possibility of using rbsB gene as a single target to inhibit plasmid-mediated horizontal gene transfer in Escherichia coli, thus we didn’t do many sgRNAs screening or using multiple sgRNAs to enhance the efficiency of CRISPRi. This can be further investigated in the follow-up studies and we have added discussion to illustrate this point.

-        It has been shown that CRISPRi imposes a polar effect on upstream and downstream genes of target gene in operon [3]. According to previous data [4], [5] rpsB is Ð° part of rbsDACBK operon. Without checking the level of expression of rbsС and rbsK, it can’t be declared that the effects observed in this research are associated with the function of rpsB.

Response: Thank you very much for your comment. The role of RbsB has not been fully investigated. The repression of rbsB gene may result in the repression of downstream genes and upstream genes in the operon, and it indeed affect many other genes expression as indicated in this study.

Based on a previous study and our RNA-seq study (as described in discussion), the role of rbsB on conjugation was investigated in this study. In this study, the repression of the rbsB gene was found directly associated with reduced conjugation rate.

-        The authors rightly noted that process of conjugation can be affected by many factors. Moreover, there is a scanty knowledge about role of RbsB on plasmid conjugation. High throughput analysis (like RNA-seq or proteomic analysis) could help to compose molecular profile of control and rpsB-downregulated strains. In this research only 8 genes were tested. The results of this study are very limited. 

[1] Goodall ECA, et al. The Essential Genome of Escherichia coli K-12. mBio. 2018

[2] Qi L.S., et al, Repurposing CRISPR as an RNA-guided platform for sequence-specific control of gene expression. Cell. 2013

[3] Peters, J.M. et al. A comprehensive, CRISPR-based functional analysis of essential genes in bacteria. Cell 2016

[4] Shimada T. et al, Involvement of the ribose operon repressor RbsR in regulation of purine nucleotide synthesis in Escherichia coli, FEMS Microbiology Letters, 2013

[5] Laikova O.N. et al, Computational analysis of the transcriptional regulation of pentose utilization systems in the gamma subdivision of Proteobacteria, FEMS Microbiology Letters, 2001

 Response: Thank you very much for your constructive comments. Actually, we chose RbsB as target based on our previous RNA-seq study (reference 30 listed in the manuscript, indicated it in the first paragraph of Discussion part). This study was to confirm the role of RbsB on conjugation. The role of RbsB need to be further investigated in the follow-up studies.

Minor comments:

-        Line 81-82,

The start of Results section confuses the reader. Please add some explanations what is plv-dCas9-101-B0, B1 and etc

 Response: Thank you very much for your kind suggestion. The detail information of all plasmids and strains have been included in Table 1. To make it more clearly, we have add explanations in results part.

-        Figure 1,

Please unify the name of the control sample (B0 or control) in the Figure 1.

 Response: Thank you very much for your kind suggestion. We have unified the name.

-        Line 87-92, Figure 1 B,

“The growth of E. coli 87 HB101 (RP4+plv-dCas9-B0) were slightly compromised in aTc concentrations up to 2 μM and 4 μM. While, the addition of 0.25 μM, 0.5 μM and 1 μM of aTc have no significant effect on the growth of the strain (Figure 1B). The dCas9 gene transcription relied on aTc induction. Thus, an appropriate aTc concentration of 1 μM was selected for triggering the CRISPRi system in the following study.”

I wonder why the authors didn’t measure the level of dcas9. If you titrate concentration of inductor, you also should measure the level of induction of target gene

 Response: Thank you very much for your comment. We have measured the expression level of dcas9 and added the results in Figure 1C and in results 2.1 part.

-        Figure 1C,

I recommend to add the expression level of rpsB in WT in order to exclude possible influence of transformation procedure and plasmid on rpsB expression.  

 Response: Thank you very much for your comment. We have added the results of expression level of rbsB in WT in Figure 1D.

-        Figure 2,

Please clarify what is Control sample in this experiment

 Response: We have added explanation of control sample in all figures’ caption.

Furthermore, we tried our best to check and revise our manuscript carefully again and again in order to improve the quality of our manuscript. Thank you and the reviewers again!

Sincerely,

Lili Li

Reviewer 2 Report

General comments:

Manuscript entitled "Inhibition of plasmid conjugation in Escherichia coli by targeting rbsB gene using CRISPRi system" describes the possibility of using the rbsB gene as a molecular target to inhibit the conjugation process in order to reduce the transmission of antibiotic resistance genes. The authors designed and conducted the study very robustly and presented the results in a clear manner. The description of research activities is concise and understandable. The research is a solid contribution to the process of combating antibiotic resistance in bacteria through a better and better understanding of the role of individual genes involved in the conjugation process and its accompanying phenomena like quorum sensing and biofilm formation. Therefore, I recommend the reviewed article for publication, but after removing certain errors, which are listed below.

Minor comments:

lines 74-75 - there is "congation", it should be "conjugation".

lines 181-182 - in the sentence "Thus, the decreased conjugation..." the fragment "...may partly due to the deceased QS..." should be changed to "...may partly result as the deceased QS..."

lines 182-183 - in the sentence "Novel mechanisms for..." the fragment "...identified widely existed..." should be changed to "...identified as widely existing..."

lines 194-196 - in the sentence "The constructed plasmid vector..." the words "be inserted" should be replaced by "provided".

line 222 - the standard code for DNA nucleotide bases is N for any of A, T, G or C, but not X.

line 293 - "Vibrio harveyi" should be in italics.

line 311 - there is "uLin", it should be "uL in".

Some sentences need rewriting for better understanding of the content presented.

Author Response

Dear reviewer,

 We wish to thank you for your effort in reviewing our manuscript and giving us an opportunity to revise our manuscript. We are also very grateful to your valuable comments in order to improve the quality of our manuscript. We have carefully reviewed and revised our manuscript according to your opinions and comments. All the revised parts are marked in red in the revised manuscript. The responses to the comments are as follows.

Minor comments:

lines 74-75 - there is "congation", it should be "conjugation".

 Response: Thank you very much for your comment. We have revised accordingly.

lines 181-182 - in the sentence "Thus, the decreased conjugation..." the fragment "...may partly due to the deceased QS..." should be changed to "...may partly result as the deceased QS..."

 Response: We have revised accordingly.

lines 182-183 - in the sentence "Novel mechanisms for..." the fragment "...identified widely existed..." should be changed to "...identified as widely existing..."

 Response: We have revised accordingly.

lines 194-196 - in the sentence "The constructed plasmid vector..." the words "be inserted" should be replaced by "provided".

 Response: We have revised accordingly.

line 222 - the standard code for DNA nucleotide bases is N for any of A, T, G or C, but not X.

 Response: We have revised accordingly.

line 293 - "Vibrio harveyi" should be in italics.

 Response: We have revised accordingly.

line 311 - there is "uLin", it should be "uL in".

 Response: We have revised accordingly.

Furthermore, we tried our best to check and revise our manuscript carefully again and again in order to improve the quality of our manuscript. Thank you and the reviewers again!

Sincerely,

Lili Li

Round 2

Reviewer 1 Report

Dear Lili Li,

In my previous report I called rbsB as rpsB several times. In all cases I meant rbsB.

Thank you for drawing my attention to reference 30 listed in your manuscript (Li, L.; Kromann, S.; Olsen, J.E.; Svenningsen, S.W.; Olsen, R.H. Insight into synergetic mechanisms of tetracycline and the selective serotonin reuptake inhibitor, sertraline, in a tetracycline-sesistant strain of Escherichia coli. J. Antibiot. 2017, 70, 944–953. doi:10.1038/ja.2017.78.)

I read it carefully and found a mention of the gene rbsB among the 50 most significant differential expressed genes in E. coli APEC_O2 treated with ST (combined sertraline and tetracycline) vs sertraline. So, expression level of rbsB was down-regulated under this condition as well as expression level other several hundred genes in this experiment. It remains unclear why you linked the reduced conjugation rate with down-regulation of rbsB gene according this study.

I still hope that you will check expression level of all genes composed rbsDECBK operon in your recombinant CRISPRi strains. 

Please check list of references, there are spelling errors.

Author Response

Dear reviewer,

Thank you very much again for your effort in reviewing our manuscript and we are very grateful to your valuable comments in order to improve the quality of our manuscript. We have carefully reviewed and revised our manuscript according to your opinions and comments. All the revised parts are highlighted in the revised manuscript. The responses to the comments are as follows.

Response: Thank you very much for your kind mention and suggestion. In our previous study, we reported the RNAseq data, and its relationship with conjugation was unpublished. We observed that the down-regulated rbsB gene was associated with reduced conjugation rate. And also in reference list 24 (Zhang, P. Y.; Xu, P.-P.; Xia, Z.-J.; Wang, J.; Xiong, J.; Li, Y. Z. Combined treatment with the antibiotics kanamycin and streptomycin promotes the conjugation of Escherichia coli. FEMS Microbiol. Lett. 2013, 348, 149–156. doi:10.1111/1574-6968.12282.), the rbsB was confirmed to have positive effects on the antibiotic induced conjugative transfer of genetic material. Based on these results, we aim to further investigated the role of rbsB gene on conjugation. We now added the results of expression level of all genes composed rbsDECBK operon in recombinant CRISPRi strains in Results (Figure S1)and Discussion.

Furthermore, we tried our best to check and revise our manuscript including the references carefully again and again in order to improve the quality of our manuscript. Thank you and the reviewers again!

Sincerely,

Lili Li